# Urinary Free Cortisol Determination and Interferences Studies Using Liquid Chromatography Coupled to Tandem Mass Spectrometry after On-Line Solid Phase Extraction Based on Turboflow^TM^ Chromatography

**DOI:** 10.3390/metabo13101063

**Published:** 2023-10-09

**Authors:** Fidéline Bonnet-Serrano, Samir Nakib, Corinne Zientek, Laurence Guignat, Jean Guibourdenche, Jerôme Bertherat, Marie-Claude Menet

**Affiliations:** 1Université Paris Cité, 75014 Paris, France; fideline.bonnet@aphp.fr (F.B.-S.); jean.guibourdenche@aphp.fr (J.G.); jerome.bertherat@aphp.fr (J.B.); 2Inserm U1016-CNRS UMR8104, 75014 Paris, France; 3Hormonology Department, Cochin Hospital, 75014 Paris, France; corinne.zientek@aphp.fr; 4Specialized Biochemistry Department, Cochin Hospital, 75014 Paris, France; samir.nakib@aphp.fr; 5Reference Center for Rare Adrenal Diseases, Endocrinology Department, Cochin Hospital, 75014 Paris, France; laurence.guignat@aphp.fr; 6Inserm U1139, 75006 Paris, France; 7Institut de Chimie Physique, CNRS UMR8000, Université Paris-Saclay, 91400 Orsay, France

**Keywords:** cortisol, steroid metabolites, Turboflow^TM^ chromatography

## Abstract

(1) A 24 h urinary free cortisol (UFF) is one of the first-line exams recommended for the diagnosis of Cushing’s syndrome. In a hospital hormonology department, this activity can exceed several hundred dosages per week. The UFF is generally determined via an immunoassay with an automate using a chemiluminescence or electrochemiluminescence detection system. To increase the cortisol concentration in the analyzed sample, the automated analysis is preceded by urine extraction, which does not prevent there from being some interferences due to other steroids with close structures. (2) This paper describes the development of on-line solid phase extraction coupled to liquid chromatography and mass spectrometry for the analysis of urinary free cortisol. The on-line extraction was based on the Turboflow^TM^ chromatography coupled to the analytical column by two valves, easily available for the laboratories. (3) The choice of the Accucore Polar Premium^®^ analytical column made it possible to avoid analytical interferences with exogenous or endogenous molecules having the same SRM transition (363 → 121) as cortisol. (4) The method was fully validated in the range of clinically relevant concentrations from the lower limit of quantification (LLOQ) to 411.75 nmol·L^−1^.

## 1. Introduction

The hormonal laboratory at Cochin Hospital works with the Reference Center for Rare Adrenal Diseases located within the Endocrinology Department of the hospital. This laboratory therefore has the opportunity to monitor a large number of patients. Urinary free cortisol (UFC) is one of the three first-line exams recommended for the diagnosis of Cushing’s syndrome [1]. It is also commonly used as the first exam in the case of a weak suspicion of cortisol excess, as a tool to eliminate Cushing’s syndrome, in frequent situations such as osteoporosis or poorly controlled diabetes [1]. Moreover, in the authentic cases of Cushing’s syndrome, whatever the etiology, UFC is the first choice analysis to monitor the effect of medical or surgical treatment, allowing the measurement of cortisol secretion integrated over 24 h. In the particular case of medical treatment, using anticortisolic drugs, a period of titration is often necessary, and UFC represents the best criteria of efficacy to adjust the treatment dose. During the titration period, UFC can be evaluated several times (3–5) a week. Taken together, these indications can represent hundreds of dosages a week, which is the usual number of UFC measurements performed each week in a center of reference for adrenal diseases such as at Cochin Hospital. The UFC is generally determined via an immunoassay with an automate using a chemiluminescence or electrochemiluminescence detection system. To increase the cortisol concentration in the sample, the automated analysis is preceded by urine extraction, which does not prevent there from being some interferences due to other steroids with close structures [2]. Liquid chromatography coupled with tandem mass spectrometry (LC-MS/MS) relies on different separation principles, orthogonal to each other, and is thus highly specific, enabling the limitation of interferences. Cortisol is first separated from other steroids and impurities via liquid chromatography before entering a triple quadrupole mass spectrometer. The latter allows the selective detection of molecules after their ionization in the source of the mass spectrometer and two successive steps of ion sorting. The first step allows the selection of the parent ion ([cortisol + H]^+^) formed in the source and the second one selects daughter ions resulting from the fragmentation of the parent ion. However, this LC-MS/MS analysis requires a prior step of cortisol extraction from the urine to eliminate some of the potential interfering substances and possibly to concentrate the urine. Two types of sample processing are described in the literature. The first one, off-line, using liquid–liquid extraction [3] or solid phase extraction (SPE) [4,5], is long because it requires the extracts to be evaporated and the dry residue to be dissolved in the mobile phase before injection into the system. The second one, offers SPE on-line with liquid chromatography [6,7]. Couchman reviewed in 2012 the use in bioanalysis of the original solid phase extraction method, Turboflow^TM^ chromatography [8,9]. The latter uses a macroporous material, which enables it to work at high mobile phase flow rate without increasing the pressure in the system (unlike classic SPE). It combines size exclusion and traditional stationary phase column chemistry to separate macromolecules, such as proteins, from smaller molecules and analytes of interest in biological fluids. This on-line extraction method is faster and requires less human intervention than the off-line one, which is very interesting in a high throughput assay context. Compared to a conventional on-line SPE column, the use of a Turboflow^TM^ column allows the sample to be loaded more quickly into the system and to have a more efficient elimination of interfering substances [8]. However, the extraction of urinary cortisol on-line, using Turboflow^TM^ technology, commonly uses a specific instrumentation called “Turboflow system”, not available in all laboratories [7].

LC-MS/MS is increasingly used for the analysis and determination of steroids in biological fluids. These complex matrices contain many endogenous and exogenous steroids, which can interfere in the assays. In particular, cortisol isomers (20α-dihydrocortisone, 20β-dihydrocortisone [7]) have the same molecular weight and daughter ions in their MS^2^ spectrum as cortisol. If the chromatographic process does not allow the separation of these compounds from cortisol, there might be an overestimation of the cortisol concentration.

We describe here the development and the analytical validation of a method with rapid handling of samples, using Turboflow^TM^ extraction on-line with ultra-high-performance liquid chromatography (UHPLC) and tandem mass spectrometry (MS/MS). The Turboflow^TM^ column was connected in a circuit of valves used in conventional SPE. The characteristics of the analytical validation met the requirements of the French accreditation committee according to the SH GTA 04 [10] guideline on bioanalytical method validation, European Medicines Agency, 21 July 2011 [11], as well as the ICH guideline Q2(R2) on validation of analytical procedures, 24 March 2022 [12]. The analytical column allowing for the best discrimination between cortisol and its isomers was selected to avoid interferences with these molecules. The validated method was applied to the high-throughput analysis of urinary cortisol. The values obtained using LC-MS/MS were compared with those obtained using the immunoassay by Liaison^®^ XL (Diasorin, Antony, France).

## 2. Materials and Methods

### 2.1. Reference Steroids and Chemicals

Formic acid was from Fluka and ammonium acetate was from Merck. Water (LCMS grade) was purchased from Carlo Erba, while methanol (Optima LCMS grade) was from Thermo Fischer (Les Ulis, France) and acetonitrile and isopropanol (HiPerSolv Chromanorm grade) were from VWR (Rosny sous Bois, France).

Cortisol and cortisol internal standard (IS), ^13^CF (cortisol-2,3,4-^13^C_3_), were purchased from Merck (Saint Quentin Fallavier, France. The commercial solutions of IS at 274 μmol·L^−1^ in methanol were successively diluted to one tenth in methanol and then to one fortieth in a mixture of water and methanol (1/1; *v/v*) to obtain the working solution at 685 nmol·L^−1^. The calibration ranges and the quality controls used for the study of the intermediary precision and the control of the assays were provided by the company BSN (Castelleone, Italy). These lyophilizates, derived from desteroidated human urine (charcoal-treated urine) and spiked with cortisol, were reconstituted in water according to the procedure provided by the manufacturer and then aliquoted. Aliquots not used at the time of reconstitution could be stored for up to 3 months at −20 °C and protected from light, according to the supplier’s procedure. The concentrations of the standards of the batch used (ref EUM06041) ranged from 0 to 1069.89 nmol·L^−1^, those of the three quality control solutions (ref EUM06051) were 6.9 mol·L^−1^ for QC1, 44.16 nmol·L^−1^ for QC3 and 209.76 nmol·L^−1^ for QC4, with an acceptability range of ±20%. Two levels (QC2 24.00 nmol·L^−1^ and QC5 334.00 nmol·L^−1^) of MassCheck^®^ Cortisol, Cortisone Urine Control kit from Chromsystem (Munich, Germany), were also used as quality controls, according to supplier recommendations.

Bétaméthasone, dexamethasone, fludrocortisone, methylprednisolone, prednisolone, prednisone and triamcinolone, used for detection of LC-MS/MS interferences and commonly prescribed at our hospital for patients concerned with urinary cortisol testing, were supplied by Merck. Urinary metabolites of steroids or steroid precursors (Androsterone, Etiocholanolone, Dehydroepiandrosterone, 5-Pregnenetriol, Pregnanediol, Tetrahydrodeoxycorticosterone, Tetrahydrodehydrocorticosterone, 5α-tetrahydrocorticosterone, 17α-hydroxypregnanolone, Pregnanetriol, Tetrahydro-11-deoxycortisol, Pregnanetriolone, 11-oxo-etiocholanolone, 11β-hydroxyandrosterone, 11-hydroxyetiocholanolone, Tetrahydrocortisone, Tetrahydrocortisol, and 5α-tetrahydrocortisol) were provided by Steraloids (https://www.steraloids.com/, accessed on 24 March 2019). The concentrations used for the study of possible interference from the latter molecules were those described in the literature (2 μg/mL [13]). 

### 2.2. Urinary Samples

All the 24 h urinary samples were collected as part of routine medical care for urinary free cortisol determination (UFF), with no additional analysis being added to the initial medical prescription.

Two urine samples were used for interference studies. The first patient took 145 mg per day of fenofibrate and the second one took 5 mg per day of prednisone.

The repeatability (intra-day) study was performed using three different pools (P1, P2 and P3) of patient urine, covering the calibration range. 

Forty-seven patient urine samples were used to compare the cortisol concentrations obtained using LC-MS/MS and using the Liaison^®^ XL immunoassay.

### 2.3. Urinary Free Cortisol Immunoassay

Before analysis via Liaison^®^ XL, free urine cortisol was extracted by mixing 2 mL of urine with 4 mL of methylene chloride (≥99.5% (*v*/*v*) stabilized, Anala^®^ NORMAPUR^®^ for analyses, VWR) for 5 min. A total of 2 mL of organic solvent was then evaporated to dryness under nitrogen. The dry residue was solubilized with 400 μL of the assay buffer provided by Diasorin [2] before cortisol quantification using the Liaison^®^ XL immunoassay. The intra-day and inter-day precision and bias of the method are within recommended specifications [2,10,11,12].

### 2.4. Mass Spectrometry (MS) 

The tandem mass spectrometry experiments were performed on a triple quadrupole mass spectrometer (Altis^®^, Thermo Fisher Scientific) with Electro Spray Ionization (ESI) in positive mode. 

The ion spray voltage was set to 3500 V, and the sheath gas, auxiliary gas and sweep gas were set to 40, 17 and 1 in abitrary units, respectively. The capillary temperature was 355 °C and vaporizer temperature was 325 °C. 

Cortisol and its IS (^13^CF) were detected using multiple reaction monitoring (MRM), with quantification and qualification transitions for each of the two molecules, i.e., 363.2 → 121.1 and 309.0 (collision energy, 26 and 18 V, respectively) and 366.2 → 124.1 and 272.00 (collision energy, 26 and 20 V, respectively). The tube lens voltage was set at 59 V for all transitions. The collision gas pressure was 1.5 mTorr. These MS parameters were optimized using cortisol and ^13^CF solutions prepared in a mixture of water and methanol (50% *v/v*), to a final concentration of 500 nmol·L^−1^. The solution was then directly infused into the ESI source with the mobile phase at its initial composition and at 500 μL/min.

### 2.5. On-Line Turboflow^TM^ Extraction and Chromatography

The chromatograph is a Vanquish Flex Binary UHPLC (Thermo Fisher Scientific) with a 6-solvent binary pump, a degasser, a thermostatically controlled automatic injector at 4 °C and 2 compartments to control column temperature at 40 °C. One Vanquish Flex UHPLC Quaternary Pump and two 6-way valves (2-Positions (2-1 and 6-1), 6-Ports, 150MPA BIO, VH-C) (Thermofisher Scientific) have been added to enable the on-line solid phase extraction.

The extraction column was a Turboflow^TM^ (TF) XL C2 (0.5 × 50 mm, ref: 953285) (Thermofisher Scientific), silica-based and suitable for the extraction of very hydrophobic compounds. Two analytical columns were used for this study, Accucore^®^ C18 (2.1 × 50 mm, 2.6 μm, ref: 17126-052130) (Thermofisher Scientific) and Accucore Polar Premium^®^ (2.1 × 50 mm, 2.6 μm, ref: 28026-052130) (Thermofisher Scientific). Both columns were based on core enhanced technology and provided high resolution separation without elevated backpressure, which is not the case in UHPLC. These columns were connected to a 6-way valve (“top valve”) as shown in Figure 1, the TF column between positions 3 and 6, and the analytical column in position 4.

The two pumps are connected to this system, the sample loading pump at position 2 of the “bottom valve” (including the injection loop) and the analytical pump at position 4 of the “bottom valve”.

The loading pump used two different solvents, an aqueous solution of formic acid (0.1%, *v/v*) and ammonium acetate (2 × 10^−3^ mol·L^−1^) (in C) to allow the sample to be loaded on the TF column, and an equi-volume mixture of water/methanol/isopropanol/acetonitrile (in D) to clean the system after elution of compounds from the TF to the analytical column.

The analytical pump also uses two different solvents, the aqueous solution (A1), identical to that used for loading sample on TF and methanol with formic acid (0.1%, *v/v*), and ammonium acetate 2 × 10^−3^ mol·L^−1^ (B1) to allow cortisol elution in gradient mode.

Table 1 describes the valve and pump configurations during process. The loading pump alternately delivers the aqueous solution (C) or the mixture of solvents (D). The sample loading on the TF column takes place at 2 mL/min during the first 30 s of the analysis. The analytical pump operates in gradient mode from 25% to 62% of B1 between 1 min and 4.5 min, with final cleaning and re-equilibration of column. Cortisol retention time is 4.23 ± 1.3% min with Accucore^®^ C18 and a total analysis time of 6 min. 

### 2.6. Sample Preparation before LC-MS/MS Analysis

To remove possible sediment, urine was centrifuged at 10,000 rpm for 10 min before analysis. A total of 10 μL of urine (centrifugated urine, calibrators and QC) were then mixed with 20 μL of the internal standard solution and 75 μL of water (LC-MS/MS grade) in an Eppendorf tube. The latter was then stirred for 10 s, left to stand for 10 min and then transferred in a vial. A total of 10μL of each mixture was then injected into the system.

## 3. Method Validation

The analytical validation criteria studied are those commonly described for the validation of assays in biological fluids. They meet the requirements of the French accreditation committee according to SH GTA 04 [10], EMA [11] and ICH [12] guidelines.

### 3.1. Recovery

The total yield of the method (extraction yield and matrix effect) was determined by comparing the area of the cortisol peak obtained after injection directly into the analytical column (without going through the extraction column) of an aqueous solution of cortisol, at 411.74 nmol·L^−1^ diluted 10 times in water (cf 2.6. sample preparation part), to the area of the cortisol peak obtained after injection of urine containing cortisol, at 411.74 nmol·L^−1^ diluted 10 times in water (calibrator 5 of the BSM calibration range) (n = 5). 

The recovery corrected by the internal standard and defined by the ratio (measured concentration/theoretical concentration) was calculated for every QC.

### 3.2. Interferences

Several molecules that could interfere in the assay of urinary free cortisol were tested. Interferences with exogenous corticoids and urinary metabolites of endogenous steroids were studied after spiking charcoal-treated urine (calibrator 0 of the BSN calibration range) with these molecules.

The known interferences of urinary metabolites of both fenofibrate and prednisone in the detection of cortisol in LC-MS/MS [14] were studied by analyzing urine of treated patients. The first patient took 145 mg per day of fenofibrate, and the second one took 5 mg per day of prednisone. 

### 3.3. Linearity and Calibration Curves

Linearity data were obtained using the program Tracefinder^TM^ (Thermo Fisher Scientific). The area ratio of each analyte to its internal standard was plotted against the corresponding concentration in nmol·L^−1^. A 1/x weighting regression was chosen to favor higher accuracy and precision at the low concentration end of the curve. For validation study, 5 calibration curves were obtained, on 5 different days, by spiking a constant concentration of internal standards (20 μL of ^13^CF at 685 nmol·L^−1^) in calibrators provided by BSN.

### 3.4. Limit of Detection (LOD) and Limit of Quantification (LOQ)

The limits of detection and quantification were determined by measuring the mean background noise (n = 10) at the cortisol retention time of the chromatograms obtained after injection of desteroidated urine. The limits were determined at signal-to-noise (S/N) ratios higher than 3 and higher than 10, for LOD and LOQ, respectively [12]. 

### 3.5. Intra-Assay and Inter-Assay Precision and Accuracy

Intra-day precision was studied by analyzing 30 times three different pools of patient urine and QC2 at 24.00 nmol·L^−1^ (provided by Chromsystem^®^), on the same day with the same operator. The inter-day precision and accuracy were studied by analyzing, on 30 different experimental days, the three QCs (6.90 nmol·L^−1^ for QC1, 44.16 nmol·L^−1^ for QC3 and 209.76 nmol·L^−1^ for QC4) of the same batch provided by BSN and QC5 (334.00 nmol·L^−1^) provided by Chromsystem^®^. The concentrations of these controls were chosen on the clinical relevance with the concentrations usually measured in the laboratory at Cochin Hospital. The precision and accuracy were also evaluated for calibration standards (n = 5).

### 3.6. Carryover

The inter-sample contamination was studied by injecting into the chromatograph 5 times the sequence HL_1_HL_2_HL_3_ of two urine samples at low (L) and high (H) concentrations. The average value of cortisol in L_1_ was compared to that of L_3_.

### 3.7. Stability

The stability of cortisol in urine is well documented in the literature, at acidic (0.5 and 1) and alkaline pH (10) and at −20 °C, 4 °C and Room Temperature (RT) [15]. 

### 3.8. Methods Comparison

Urinary free cortisol of 47 patients were assayed in parallel using LC-MS/MS and Liaison^®^ XL. The linear regression allowed us to assess the agreement between these two series of measurements. 

## 4. Results and Discussion

### 4.1. Extraction Method Development

Three silica-based Turboflow^TM^ extraction columns were tested, Turboflow^TM^ XL C2, Turboflow^TM^ XL C8 and Turboflow^TM^ XL C18 (0.5 × 50 mm, ref: 953285, 953282, 953280, respectively, Thermofisher Scientific), differing in the hydrophobicity of their stationary phase. For these experiments, the analytical column was Accucore^®^ C18 (2.1 × 50 mm, 2.6 μm). The cortisol concentration of the solution injected was 411.74 nmol·L^−1^ before dilution in water (cf 2.6. sample preparation part). As the stationary phases based on octyl and octadecyl grafted silica were very hydrophobic, the transfer of cortisol from the TF column to the analytical column and its elution from the analytical column required a high initial percentage of methanol (40% of B1). The resulting peaks were very asymmetrical and large (Appendix A). This was not the case for the extraction with the column based on ethyl grafted silica. When the C2 grafted phase was used, the retention was lower. A lower percentage of B1 (25%) made it possible to obtain the elution of cortisol from the C2 grafted silica (compared to 40% for the C8 and C18 grafted silicas). The peak obtained was much narrower and symmetrical at a reproducible retention time of 4.23 min ± 1.3%.

The chromatogram obtained with the C2XL extraction column is shown Appendix A. It is not on the same scale as the others. Indeed, the peak obtained is narrower and therefore higher. The total yield of the method (extraction yield and matrix effect) was measured at 68.2% (CV 7%, n = 5). The yields (n = 16) corrected by the internal standard were close to 100% within + 15%, with 105.8% (CV 7.9%), 88.3% (CV 5.7%), 101.9% (CV 6.7%) and 99.5% (CV 3.5%), respectively, for QC1, QC3, QC4 and QC5.

### 4.2. Analytical Validation

#### 4.2.1. Interferences—Selectivity

Exogeneous steroids and urinary metabolites of endogenous steroids listed in Section 2.1. The “Reference steroids and chemicals” section did not interfere with urinary cortisol analysis via LC-MS/MS, whatever the analytical column used, Accucore^®^ C18 or Accucore Polar Premium^®^. The solutions containing these molecules did not give any peak at the retention times of cortisol and ^13^CF (Appendix A). 

However, during routine urinary cortisol assays using LC-MS/MS, interference problems with other compounds were observed. The interference already described with fenofibrate [14] for the cortisol assay via LC-MS/MS was studied using urine from a patient treated with this drug. The major metabolites of fenofibrate are fenofibric acid, its reduced form and their glucurono-conjugated derivatives [16]. These molecules have lower or much higher molecular weights (for the glucuronides) than that of fenobibrate, 360.1 g/mol.

Figure 2A shows that fenofibrate has a Cl atom in its structure. The isotopic pattern of its molecular ion [C_20_H_21_ClO_4_ + H]^+^ therefore has two peaks at *m/z* 361.1 and 363.1 corresponding to the ions [C_20_H_21_^35^ClO_4_ + H]^+^ and [C_20_H_21_^37^ClO_4_ + H]^+^, respectively. The intensity of the peak at *m/z* 363.1 is high (1/3 of that of the ion at *m/z* 361.1), which is confirmed by the spectrum available in MassBank [17]. This library also gives the MS^2^ fragmentation spectrum of [C_20_H_21_ClO_4_ + H]^+^ consisting of three daughter ions, [C_7_H_4_O_2_ + H]^+^ at *m/z* 121.03, [C_7_H_4_ClO + H]^+^ at *m/z* 138.9 and [C_13_H_10_ClO_2_ + H ]^+^ at *m/z* 233.0 (Figure 2A). When cortisol and fenofibrate are eluted at the same retention time, the daughter ion [C_7_H_4_O_2_ + H]^+^ causes interference with the 363 → 121 transition used for the cortisol assay. Some authors [14] propose to eliminate this interference by changing the transitions used for the cortisol assay, replacing 363 → 121 with 363 → 97 (Figure 3A). However, 363 → 121 is the most sensitive transition for the cortisol assay. In addition, a modification of the cortisol assay method for a small number of samples (patients with fenofibrate) can be a problem during routine urinary cortisol assays in a medical biology laboratory.

Figure 2B shows the chromatograms obtained with the Accucore^®^ C18 column after injection of urine from a patient treated with fenobibrate. There is a chromatographic peak at 4.27 min for the 363 → 121 transition but no peak emerging from the background noise for the 363 → 309 transition. The chromatogram of the 366 → 124 transition shows the elution of ^13^CF, ^13^C-labeled cortisol whose retention time is identical to that of unlabeled cortisol. This peak has the same retention time as the peak recorded for the 363 → 121 transition.

Figure 2C shows the chromatogram obtained after analyzing the urine of a patient treated with fenofibrate using the Accucore Polar Premium^®^ column. It shows a separation of the two peaks, that of cortisol (here represented by cortisol labeled with ^13^C (^13^CF) with the same retention time as unlabeled cortisol), which is eluted at 4.0 min, and that of fenofibrate, eluted at 4.26 min. This result confirms that the patient’s urine contains very little cortisol; the peak at 4.0 min is not detected for the transition 363 → 121. The use of this column therefore makes it possible to elute cortisol (4.0 min) and fenofibrate (4.26 min) at different retention times. It allows the use of the transition 363 → 121 (as a quantification transition), whether the patient is treated or not by fenofibrate.

Another interference was observed with prednisone metabolites, which are cortisol isomers (Figure 3A). Indeed, prednisone is reduced in the liver to tetrahydrogened derivatives (5α,20β-tetrahydroprednisone, 5β,20β-tetrahydroprednisone, 5α,20α-tetrahydroprednisone, and 5β,20α-tetrahydroprednisone) [18,19,20,21,22,23,24]. It can also be reversibly metabolized to prednisolone (Figure 3A), which is then reduced to 20β-dihydroprednisolone and 20α-dihydroprednisolone [25,26,27,28,29]. All these molecules are isomers of cortisol and have the same molecular weight. Some of them interfere with the assay of cortisol using LC-MS/MS with the Accucore^®^ C18 column, as shown in a patient treated with prednisone (Figure 3B).

Figure 3B was obtained using an Accucore^®^ C18 column. The retention time of cortisol is 4.27 min (retention time of ^13^CF given by the plot of 366 → 124). Figure 3B highlights that there is a peak at the cortisol retention time on the 363 → 121 plot, at the same retention time as ^13^CF. With this column, there is therefore coelution of cortisol with a compound that appears in the serum after prednisone administration. The cortisol concentration determined using the Accucore^®^ C18 column is therefore incorrect in this case.

Once again, the Accucore Polar Premium^®^ column allowed a better separation of cortisol from interfering substances compared to the Accucore^®^ C18 column. The Accucore Polar Premium^®^ column had previously been used by Sanchez-Guijo et al. [7] for the separation of cortisol from isobaric steroids, sharing the same transitions with cortisol, 20α-dihydrocortisone and 20β-dihydrocortisone. We showed here that it could also efficiently separate cortisol from prednisone metabolites in urine. This could be explained by the nature of the stationary phase contained in the Accucore Polar Premium^®^ column, which has an integrated amide function offering a better chromatographic selectivity than conventional C18 columns.

Looking at the transition 363 → 121 (Figure 3C), it was possible to see that the intensity of the cortisol peak at 4.01 min was low, corresponding to a low cortisol concentration in the patient urine, which is expected in patients undergoing corticotherapy. The four interferents detected at 3.07, 3.49, 3.65 and 3.86 min could correspond to prednisone metabolites described in Figure 3A. The first two peaks at 3.07 and 3.49 min, respectively, are the least retained by the column and therefore correspond to the most hydrophilic compounds. These compounds could be derivatives of prednisolone with 4 OH groups (dehydroprednisolone). The two compounds eluted later might correspond to derivatives of prednisone (tetrahydroprednisone) with only 3 OH groups in their structure.

This hypothesis can be confirmed by the results obtained for the transition 363 → 309. The fragment ion 309 corresponds to three successive losses of water molecules [26]. These losses are very easy and occur when the molecule has alcohol functions. Prednisone derivatives have three OH groups, which promotes the formation of fragment ions at *m/z* 309, unlike prednisolone derivatives, which have four OH groups. For the latter, a very small peak is detected for the transition 363 → 309 at 3.47 min, with no peak at 3.07 min. 

Finally, the Accucore Polar Premium^®^ column allowed the separation of some but not all isomers since there are only two peaks for the potential four tetrahydrogen derivatives, 5α,20β-tetrahydroprednisone, 5β,20β-tetrahydroprednisone, 5α,20α-tetrahydroprednisone and 5β,20α-tetrahydroprednisone.

#### 4.2.2. Linearity and Calibration Curves 

A straight line was fitted to the data points using the least-square regression analysis. All five standard calibration curves analyzed are linear with a determination coefficient r^2^ from 0.9975 to 0.9994. Moreover, the standard line intercept is not significantly different from 0 at the 95% confidence interval (0.0042/0.0073 < 2.069 (t_95%_, 28)) and the slope is significantly different from 0 at the 95% confidence interval (6 × 10^−4^/1.4 × 10^−5^ > 2.069).

Data for the linearity of the method are shown in Table 2, which gathers the precision (CV%) and the average deviation from the theoretical value (Bias%). These results satisfy the current criteria for bio-analytical methods [10,11,12], the precision was <15.0% and the deviation from nominal concentration was within +15%.

#### 4.2.3. LOQ and LOD

The background noise measurement was performed after the injection of 10 charcoal-treated urine samples provided by BSN. The values of LOD and LOQ are 0.55 nmol·L^−1^ and 1.65 nmol·L^−1^, respectively. 

These values are comparable to those encountered in the literature (2.7 nmol·L^−1^ or 1 ng/mL) [7]. However, the quantity of urine treated is much lower, 10 μL for our method against 100 μL [7] or even 2 mL [30], with a quantity of cortisol injected at the LLOQ of 1.65 fmol in our method versus 54 fmol for [7].

#### 4.2.4. Intra-Assay and Inter-Assay Precision and Accuracy 

The results for intra- and inter-assay variations of all P and QC samples (Table 3) satisfy the current criteria for bio-analytical methods, as the precision was <15% and the deviation from nominal concentration within +15%. The bias could not be calculated for the urinary pools since their theoretical concentration is unknown. The deviation from nominal concentration was 0.5% for QC2, i.e., within +15% as recommended by guidelines [10,11,12].

The inter-day precision study was performed over six months, during which we can perform up to 4000 injections, while maintaining good chromatography and good precision and accuracy.

#### 4.2.5. Carryover

The inter-sample contamination was studied by injecting into the chromatograph 5 times the sequence HL_1_HL_2_HL_3_ of two urine samples at low (L) and high (H) concentrations. The average value of cortisol in L_1_ was compared to that of L_3_.

The average value of cortisol in H_1_, H_2_ and H_3_ samples was 1144.73 in arbitrary units (CV 2.60%). The average value of cortisol in L_1_, L_2_ and L_3_ samples was 8.49 in arbitrary units (% CV 3.07). The average value of cortisol and standard deviations were 8.72 and 0.29%, and 8.40 and 0.13%, for L_1_ and L_3_, respectively. The average cortisol values between L_1_ and L_3_ are not significantly different (1.94 < 2.77 (t_95_^%^, 4). There is no cross contamination between samples at the time of injection.

#### 4.2.6. Methods Comparison between Liaison^®^ XL and Proposed Method

Figure 4 shows the correlation between the urinary free cortisol concentration values determined using our LC-MS/MS method and using the Liaison^®^ XL (Diasorin) immunoassay. To perform this correlation, we used clinically relevant cortisol concentrations, lower than 255 nmol·L^−1^ using LC-MS/MS (n = 32). Appendix A shows the correlation obtained for n = 47.

Concentration values higher than the highest concentration of the calibration range were determined after adequate dilution in water for LC-MS/MS and in the buffer recommended by the supplier for Liaison^®^ XL. The correlation line has a correlation coefficient of 0.7177. The slope of the equation (0.5041), lower than 1, highlights that the cortisol concentrations determined using Liaison^®^ XL are higher than the concentrations determined using LC-MS/MS as expected [31], which can be explained by the very high specificity of LC-MS/MS unlike methods using antibodies.

Moreover, UFC determination via immunoanalysis most often requires a preliminary step of extraction (for example, liquid–liquid extraction with dichloromethane), which is particularly time-consuming. Our method using on-line SPE represents a very convenient alternative for high-throughput analysis. Another issue with immunoanalysis is the cross-reactivity of the anti-cortisol antibody with other endogenous or exogenous steroids, such as prednisone and prednisolone. As illustrated, the Liaison^®^ XL cortisol immunoassay from Diasorin, previously used for UFC determination in our laboratory, presents a 12.6% cross-reactivity with prednisolone [2]. Our approach has the big advantage of allowing the separation of cortisol from exogenous corticoids. Indeed, the choice of the Accucore Polar Premium^®^ analytical column made it possible to avoid analytical interferences with both endogenous and exogenous molecules and metabolites sharing the same SRM transition (363 → 121) as cortisol.

A CFU assay costs EUR 3.5 using the Liaison^®^ XL immuno analysis (cost of kits) and EUR 1 with our SPE on-line with LC-MS/MS method (cost of standard and control kits, extraction and analytical columns, and solvents). This low price can be partly explained by the low volume of standards and controls (10 μL) used for the assay.

The good correlation (Figure 4) between the two methods allowed us to calculate adapted reference values for urinary free cortisol in LC-MS/MS. Applying the regression line equation on the cut-off proposed by the supplier Diasorin [2] (<82.5 μg/24H, i.e., 227.6 nmol/24H, n = 50), we found a new cut-off of 126.0 nmol/24H in LC-MS/MS. This value is consistent with those proposed in the literature for urinary free cortisol determined using LC-MS/MS, up to 186 nmol/24H (20 men and 50 women) [32] and up to 166 nmol/24H (83 men) and 119 nmol/24H (104 women) [3]. The value we determined needs to be refined using a larger cohort of patients and by specifically determining normal levels in men and women.

#### 4.2.7. Methods Comparison between LC-MS/MS Methods

Examples of LC-MS/MS methods, described in the literature, are gathered Table 4. They analyze the cortisol contained in different matrices (urine, plasma, saliva and hair). The first extraction step is off-line or on-line before analysis using LC-MS/MS. Liquid–liquid extraction, solid phase extraction and precipitation can be used off-line for the analysis of urinary, plasma and salivary cortisol.

Liquid–liquid extraction, with solvents immiscible with biological media, uses a large volume of urine (1 mL), which makes it possible to obtain a low LOQ comparable to that obtained for the proposed method (1.4 nmol·L^−1^ (or 0.5 ng/mL) [33] versus 1.65 nmol·L^−1^). This extraction method takes much longer than the proposed method because it requires the organic solvents’ evaporation after extraction. The chromatographic method is short and comparable to the proposed method (cortisol retention time, 1.5 or 5.3 min and 4.3 min, respectively), but the authors do not discuss possible interfering molecules.

Wear et al. [34] and Persichilli et al. [35] describe precipitation methods (using methanol or trichloroacetic acid) with LOQ values higher than that of the proposed method (minimum of 2.5 nmol·L^−1^ versus 1.65 nmol·L^−1^). A total of 50 μL of treated urine was injected into the system [34]. The authors did not detect any ion suppression or interferences. However, precipitation extraction takes longer than the proposed method. It requires centrifugation of the treated samples before injection into the chromatograph.

Solid phase extraction using Oasis^®^ HLB is widely used for the extraction of steroids from biological fluids. As a liquid–liquid extraction, it requires the solvent to be evaporated after extraction and just before injection into the chromatograph [4,36]. The LOQ values obtained are comparable to that of the proposed method (from 1 nmol·L^−1^ to 3.75 nmol·L^−1^ versus 1.65 nmol·L^−1^), but it uses a higher initial urine volume (from 50 μL to 1 mL versus 10 μL).

Cortisol extraction on-line with LC-MS/MS is used for the analysis of different biological materials as described in Table 4, such as hair, saliva and urine. It is difficult to compare the performance of such a method between hair and urine analyses. Regarding the total analysis time, before injection into the chromatograph, the hair must undergo a long treatment to extract free cortisol, which is not the case with urine. Additionally, urine and hair extracts do not contain the same molecules that could interfere or change (decrease or enhance) ionization in the mass spectrometer, which would change the LOQ values.

Urine extractions on-line with LC-MS/MS and described in Table 4 use a higher volume of urine than that used in the proposed method (from 50 μL to 500 μL of urine versus 10 μL) with similar or even higher LOQ values. The quantities of cortisol detected at the LOQ are therefore lower in the proposed method (cf 3.3.3 LOD and LOQ).

Two studies using extraction on-line with LC-MS/MS analysis [7,14] studied cortisol interference with fenofibrate [14] and cortisol isomers [7]. These molecules have the same retention time as cortisol. Kushnir et al. [14] suggest eliminating the interference by changing the SRM transition used for the analysis (from 363 → 121 to 363 → 97), which is not easy in a high-throughput context and requires reinjecting the sample under new setting conditions of the mass spectrometer. The method described here avoids this drawback by using a chromatographic column that separates cortisol from fenofibrate. Sanchez-Guijo et al. [7] described an on-line extraction method with a TurboFlow^TM^ column followed by chromatography allowing for the separation of cortisol from its isomers. But the on-line extraction requires specific equipment, the “TurboFlow^®^ System”, which is not the case in the method proposed.

## 5. Conclusions

This paper describes the development of a new method for the analysis of UFC, based on on-line solid phase extraction with liquid chromatography and mass spectrometry and currently used in the hormonal laboratory of Cochin Hospital instead of the old immunoanalysis method using Liaison^®^ XL.

The on-line extraction LC-MS/MS analysis described in this paper beneficiates from the selectivity of the Accucore Polar Premium^®^ analytical column and the efficiency of the System Turboflow^TM^ with easy implementation, since the extraction column was coupled to the analytical column by two valves easily available at the laboratory level. The method was fully validated in the range of clinically relevant concentrations, from 1.65 nmol·L^−1^ to 411.75 nmol·L^−1^.

## Figures and Tables

**Figure 1 metabolites-13-01063-f001:**
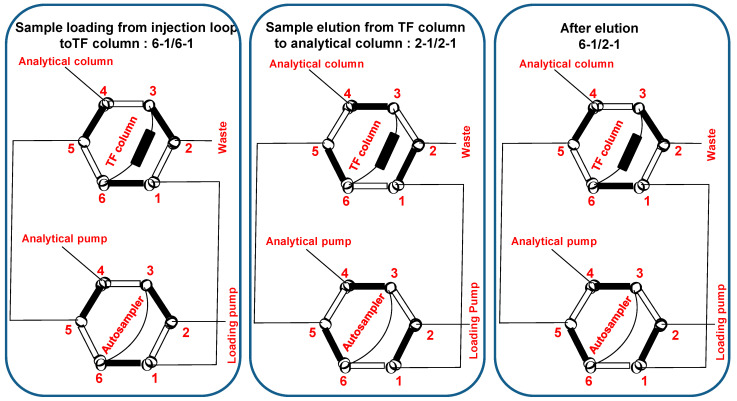
Column and pump connections. The mobile phases pass through the black tubes. The valves at the top of the figure represent the valve connected to the analytical column and to the TF column in the three conditions, sample loading, sample elution and after elution. The valves at the bottom of the figure represent the valve connected to the analytical pump, to the loading pump and to the autosampler in the three conditions, sample loading, sample elution and after elution. The connections are numbered from 1 to 6. Sample loading: top and bottom valves at 6-1 positions. Sample elution: top and bottom valves at 2-1 positions, the mobile phase from the analytical pump allows the cortisol elution from the TF column and from the analytical column. After elution (TF column rinsing), top valve at 6-1 position, and bottom valve at 2-1 position.

**Figure 2 metabolites-13-01063-f002:**
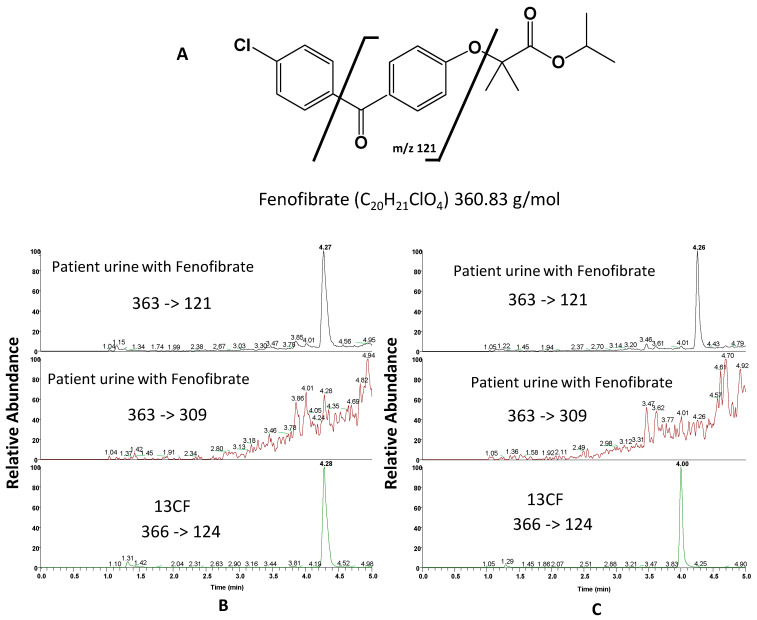
(**A**) Fenofibrate and daughter ion *m/z* 121 [17]. (**B**) Accucore^®^ C18 column and from top to bottom: chromatograms obtained after injection of urine from a patient treated with fenobibrate (transition of cortisol quantification (363 → 121) and qualification (363 → 309) spiked with a ^13^CF solution (366 → 124). (**C**) Accucore Polar Premium^®^ column and from top to bottom: chromatograms obtained after injection of patient urine treated with fenobibrate (transition of quantification and qualification) spiked with a solution of ^13^CF.

**Figure 3 metabolites-13-01063-f003:**
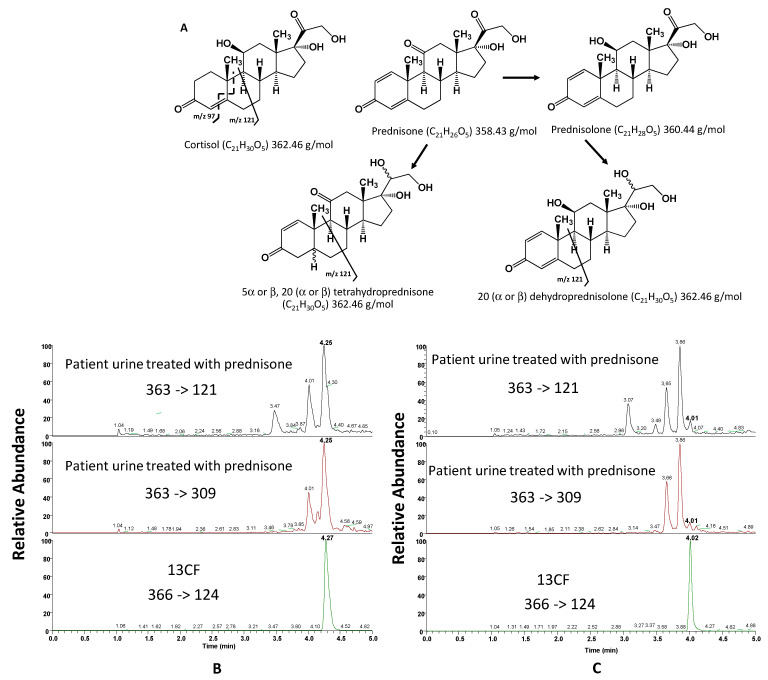
(**A**) Cortisol isomers, metabolites of prednisone and prednisolone likely to appear in patients treated with prednisone. (**B**) Accucore^®^ C18 column and from top to bottom: chromatograms obtained after injection of urine from a patient treated with prednisone (transitions for quantification (363 → 121) and qualification (363 → 309) of cortisol) spiked with a ^13^CF solution (366 → 124). (**C**) Accucore Polar Premium^®^ column and from top to bottom: chromatograms obtained after injection of urine of patient treated with prednisone (transitions of quantification and qualification for cortisol) spiked with a solution of ^13^CF.

**Figure 4 metabolites-13-01063-f004:**
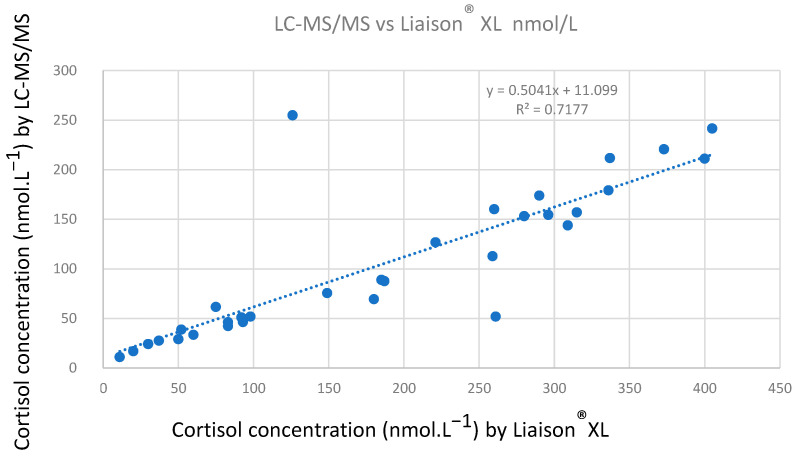
Correlation between urinary free cortisol concentration values determined using LC-MS/MS and Liaison^®^ XL (n = 32).

**Table 1 metabolites-13-01063-t001:** Valve and pump configurations.

		TF Column/Tubing	Analytical Column	Loading Pump		Analytical Pump	
Time (min)	Valves			Flow Rate (mL/min)	Solvent	Flow Rate (mL/min)	Solvent
	6-1, 6-1				100% C	0.5	25% B1
0		Sample loading		2	100% C	0.5	25% B1
0.5	2-1, 2-1			2	100% C	0.5	25% B1
0.6		Cortisol transfer/Cleaning tubing with D	Cortisol transfer	2	100% D	0.5	25% B1
1			Elution gradient	2	100% D	0.5	25% B1
2.7				0.1	100% D	0.5	
2.9				2	100% D	0.5	
3.6		Cleaning tubing with C		2	100% C	0.5	
4.5				2	100% C	0.5	62% B1
4.6	6-1, 2-1		Cleaning	2	100% C	0.5	98% B1
5		Equilibration with C	Equilibration with 25% B1	2	100% C	0.5	25% B1
6	6-1, 6-1			2	100% C	0.5	25% B1

**Table 2 metabolites-13-01063-t002:** Linearity and calibration curves (n = 5).

	Theoretical Values (nmol·L^−1^)	Average nmol·L^−1^	Standard Deviation	CV (%)	Bias (%)
S1	6.65	6.97	0.56	8.00	4.44
S2	29.15	29.7	1.18	3.99	1.86
S3	132.87	122.76	4.32	3.46	−5.83
S4	89.50	86.44	4.28	4.95	−3.42
S5	411.74	409.63	11.88	3.01	−4.21
S6	1069.89	1081.24	9.16	0.84	2.26

**Table 3 metabolites-13-01063-t003:** (**a**) Intra-assay, n = 30. QC2 Chromsystem control, and Px pools of patient urines. (**b**) Inter-assay (n = 30). QC1, QC3 and QC4 BSN control, and QC5 Chromsystem control.

(**a**)				
**Pool**	**Average (nmol·L^−1^)**	**Standard Deviation**	**CV (%)**	
QC2	23.9	1.5	6.2	
P1	97.9	2.1	2.1	
P2	361.4	11.7	3.2	
P3	784.1	17.0	2.2	
(**b**)				
	**QC1**	**QC3**	**QC4**	**QC5**
Target nmol·L^−1^	6.90	44.16	209.76	334.00
Average nmol·L^−1^	7.3	39.0	206.1	332.4
CV (%)	5.8	6.9	7.8	3.4
Bias (%)	5.2	−11.6	−1.8	−0.5

**Table 4 metabolites-13-01063-t004:** Extraction methods on-line or off-line with LC-MS/MS, described in the literature.

Reference	Sample	Extraction Mode	Cortisol Retention Time Injection Interval	LOQ or LCCR *
[33]	Plasma (250 μL)/Urine (1 mL)	Liquid–liquid extraction using 3 mL of ethyl acetate	5.3 min	LOQ 0.5 ng/mL
[3]	Urine (500 μL)	Liquid–liquid extraction using 4.5 mL of methylene chloride	1.5 min	LCCR 7 nmol·L^−1^
[34]	Urine (100 μL)	Injection volume: 50 μL Precipitation with trichloroacetic acid and centrifugation	2 min	LOQ 2.5 nmol·L^−1^
[35]	Urine (500 μL)	Precipitation with methanol and filtration	6.84 min	LCCR 7 nmol·L^−1^
[30]	Urine (125 μL)	Injection volume: 20 μL Solid phase extraction with Oasis^®^ HLB	13 min	LOQ 0.3 ng/mL
[36]	Urine (1 mL)	Solid phase extraction with Oasis^®^ HLB	2.6 min	LOQ 2 nmol·L^−1^
[4]	Acidified plasma (250 μL), urine (50 μL), saliva (250 μL), ultrafiltrate of plasma (250 μL)	Solid phase extraction with Oasis^®^ HLB	1.46 min	LOQ in plasma 3.75 nmol·L^−1^
[37]	Urine (500 μL) diluted by 10	On-line extraction (Zorbax Extend-C18 cartridge)	4.4 min	5 nmol·L^−1^
[38]	Saliva (200 μL)	Injection volume: 50 μL On-line extraction (Phenomenex C8)	3 min	2 nmol·L^−1^
[39]	Hair (8 mg) in 1.8 mL of MeOH	1.6 mL evaporated, residue dissolved in 500 μL Injection volume 50 μL On-line extraction with cartridge (2 × 4 mm, C18, Phenomenex^®^)	6.3 min	0.8 pg/mg
[40]	Hair (50 mg) in 2 mL of MeOH	50 μL of supernatant On-line extraction with Restricted Acess Material phase (Lichrospher^®^ RP-8-ADS (25 μm, 24 mm × 4 mm)	11.5 min	2 pg/mg
[14]	Urine (500 μL)	Injection volume: 200 μL On-line extraction (C18 Phenomenex)	Injection interval: 8 min	LCCR 10 μg/L
[6]	Urine (50 μL) diluted by 20	Injection volume (diluted urine): 60 μL On-line extraction (Poros)	4.07 min	LCCR 5 ng/mL
[7]	Urine (100 μL)	Injection volume: 20 μL On-line extraction (Turbulent Flow Chromatography: C18 HTLC column)	5.70 min	LOQ 1 ng/mL
Proposed method	Urine (10 μL) diluted by 10	Injection volume: 10 μL On-line extraction (Turboflow^TM^ XL C2)	4.3 min	LOQ 1.65 nmol·L^−1^

* LCCR: the lowest concentration of calibration range.

## Data Availability

Chromatograms for interference studies and analytical validation are provided in the publication. Raw data for correlation are provided in Appendix A.

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
