# Peer review of "Urinary Free Cortisol Determination and Interferences Studies Using Liquid Chromatography Coupled to Tandem Mass Spectrometry after On-Line Solid Phase Extraction Based on TurboflowTM Chromatography"

_metabolites, 2023, doi:10.3390/metabo13101063_

Round 1
Reviewer 1 Report
The study is interesting, but the problem is the application in patients of this method as an alternative to immunoassay determination.
The statement that a few thousand assays per year are performed in endocrinology departments is not true. Cortisol assessment is performed either for suspect s of Cushing's, Addison's,to evaluate a tertiary hypocortisolism or other pathologies and the cases are not so frequent.
Authors should demonstrate that prednisone therapy shows elevated cortisol values by traditional methods,
Authors should specify the cost of such method.
Figure 4 shows that the proposed method gives similar results with Liaison KL
Translated with DeepL
Reviewer 2 Report
To the authors:
1. General comments:
The manuscript “Urinary free cortisol determination and interferences studies by liquid chromatography coupled to tandem mass spectrometry after online solid phase extraction based on TurboflowTM chromatography” is a very interesting original article that describes the method for the accurate analysis of cortisol (a very important metabolite) in urine by MRM. The text is well-written and easy to follow. However, there are a few important comments that I consider should be corrected before publication:
2. Specific comments for revision: b) major.
1. For M&M and later in results, please for concentration units unify by employing molarity instead of ppb or ppm. It is quite odd to see sometimes molarity and others ppb.
2. Please discuss why the best SPE column for cortisol the C2XL was better than the C8 or C18 and was more retained, if according to the structure it would be expected to be opposite.
Minor comments:
1. Line 32. Define LLOQ.
2. Line 44. Change LC MS/MS for LC-MS/MS throughout the text.
3. Line 55. Which method is tedious? SPE or LLE?
4. Line 59. Once SPE is defined use the abbreviation.
5. Line 73. Add some examples of cortisol isomers.
6. Line 79. Use only MS/MS as it was defined before.
7. Line 88. Include that Liaison is the immunoassay.
8. Line 148. Use the abbreviation IS
9. Line 155. Define ESI.
10. Figure 1. Please explain the difference between the up and bottom graphs.
11. Lines 225-226 are repeated.
12. Line 236. Include the abbreviations for LOD and LOQ
13. Line 254, include the pH values
14. Use italics for m/z throughout the text.
15. Figure 4, Fig 2S. Add the X- and Y-axis

The text should be corrected by a native speaker, as there are some sentences difficult to understand.
Reviewer 3 Report
Publication is interesting and worth publishing, it will certainly find interest among readers. All the research was done properly. However, the authors should make some small corrections first:
Line 274-257: what doest it mean: ‘the peak is more efficient’?
Line 287: please show all of these results in supplement
The authors need to add a chapter with a description of how their methods compare with other methods used to extract this compound. Authors should include in this chapter a table, a description and a clear indication of the advantages and disadvantages of the developed approach.
The novelty of this research should also be very clearly indicated. Many papers may be already found in the literature about SPE LC-MS3 methods for extraction and determination of cortisol (e.g. doi: 10.1016/j.talanta.2014.11.034, doi.org/10.1007/s11696-018-0560-1, dx.doi.org/10.1007/s11696-018-0560-1, doi.org/10.1258/acb.2009.009053, and many others) whats the novelty here?
Round 2
Reviewer 2 Report
To the authors:
Thanks to the authors for the changes, the manuscript has improved greatly. I recommend it for publication after the correction of minor comments:
1. Figure 4. The values of the equation of the curve and the R2 are with commas instead of dots. Please review and change accordingly.
2. Table 4 has been mentioned in the text as IV, please unify.
3. Line 160. After “electrospray ionization” abbreviate as ESI, and then use this in line 172.
4. The conclusion is now very long, I suggest changing Lines 530 to 584 and Table 4 to the results and discussion section.
Author Response
Dear reviewer,
We thank you for your comments.
You will find in the file the minor modifications (highlighted in yellow, lines 160 (ESI), 172 (ESI), 518 (Table 4).
The conclusion part has been reduced. A final part 3.3.7. "Methods comparison between LC-MS/MS methods" was added in Results and discussion.
I added "Supplementary materials" at the end of the publication.
Best regards.
Marie-Claude Menet
